# Seminal Plasma Antioxidants Are Related to Sperm Cryotolerance in the Horse

**DOI:** 10.3390/antiox11071279

**Published:** 2022-06-28

**Authors:** Jaime Catalán, Iván Yánez-Ortiz, Asta Tvarijonaviciute, Luis Guillermo González-Aróstegui, Camila P. Rubio, Isabel Barranco, Marc Yeste, Jordi Miró

**Affiliations:** 1Equine Reproduction Service, Department of Animal Medicine and Surgery, Faculty of Veterinary Sciences, Autonomous University of Barcelona, ES-08193 Cerdanyola del Vallès, Spain; dr.jcatalan@gmail.com (J.C.); ivan.yanez22@gmail.com (I.Y.-O.); 2Biotechnology of Animal and Human Reproduction (TechnoSperm), Institute of Food and Agricultural Technology, University of Girona, ES-17003 Girona, Spain; 3Unit of Cell Biology, Department of Biology, Faculty of Sciences, University of Girona, ES-17003 Girona, Spain; 4Faculty of Veterinary Medicine, University of Teramo, Loc. Piano d’Accio, IT-64100 Teramo, Italy; 5Department of Animal Medicine and Surgery, Faculty of Veterinary Medicine, University of Murcia, ES-30100 Murcia, Spain; asta@um.es (A.T.); luisgarostegui@gmail.com (L.G.G.-A.); 6Interdisciplinary Laboratory of Clinical Analysis Interlab-UMU, Faculty of Veterinary Medicine, Regional Campus of International Excellence ‘Campus Mare Nostrum’, University of Murcia, ES-30100 Murcia, Spain; camila.peres@uab.cat; 7Department of Animal and Food Science, Faculty of Veterinary Sciences, Autonomous University of Barcelona, ES-08193 Cerdanyola del Vallès, Spain; 8Department of Veterinary Medical Sciences, University of Bologna, IT-40064 Ozzano dell’Emilia, Italy; isabel.barranco@unibo.it; 9Catalan Institution for Research and Advanced Studies (ICREA), ES-08010 Barcelona, Spain

**Keywords:** antioxidants, cryopreservation, horse, oxidative stress, reactive oxygen species, seminal plasma, sperm

## Abstract

The objective of this study was to determine the relationship of enzymatic (superoxide dismutase, SOD; glutathione peroxidase, GPX; catalase, CAT; and paraoxonase type 1, PON1) and non-enzymatic antioxidants (measured in terms of: Trolox equivalent antioxidant capacity, TEAC; cupric-reducing antioxidant capacity, CUPRAC; and ferric-reducing ability of plasma, FRAP), as well as the oxidative stress index (OSI) in seminal plasma (SP) with the resilience of horse sperm to freeze-thawing. Twenty-one ejaculates (one per individual) were collected and split into two aliquots: the first was used to harvest the SP and assess the activity levels of antioxidants and the OSI, and the second one was cryopreserved. The following post-thaw sperm quality parameters were evaluated: sperm motility, plasma membrane and acrosome integrity, mitochondrial membrane potential, intracellular levels of reactive oxygen species (ROS), and plasma membrane lipid disorder. Based on post-thaw total motility (TM) and plasma membrane integrity (SYBR14^+^/PI^−^), ejaculates were hierarchically (*p* < 0.001) clustered into two groups of good (GFE) and poor (PFE) freezability. The SP activity levels of PON1, SOD, and TEAC were higher (*p* < 0.05) in GFE than in PFE, showing a positive relationship (*p* < 0.05) with some sperm motility parameters and with plasma membrane (PON1 and TEAC) and acrosome (SOD and TEAC) integrity. In contrast, OSI was higher (*p* < 0.05) in the SP of PFE than in that of GFE, and was negatively correlated (*p* < 0.05) to some sperm motility parameters and to plasma membrane and acrosome integrity, and positively (*p* < 0.05) to the percentage of viable sperm with high plasma membrane lipid disorder. In conclusion, enzymatic (PON1 and SOD) and non-enzymatic (TEAC) antioxidants of SP are related to horse sperm cryotolerance. In addition, our results suggest that PON1 could be one of the main antioxidant enzymes involved in the control of ROS in this species. Further investigation is needed to confirm the potential use of these SP-antioxidants and OSI to predict sperm cryotolerance in horses.

## 1. Introduction

Sperm cryopreservation has acquired great importance in recent decades, due to the benefits offered by its use [1,2,3]. Freeze-thawing procedures, however, impact negatively on sperm motility, morphology, functionality, and survival [4,5], as well as on their fertilizing capacity [6]; this leads to lower pregnancy rates after artificial insemination compared to fresh or cooled semen [7,8]. During cryopreservation, a stressful environment is generated for sperm [9], mainly associated with an excess of reactive oxygen species (ROS) [6] that overwhelms the sperm antioxidant defense capacity. This situation drives the oxidative/antioxidant balance to shift towards the oxidative state [10,11]. In this circumstance, oxidative stress underlies several sperm injuries at the level of plasma membrane (lipoperoxidation) [12,13,14] and DNA (fragmentation) [11,15,16], and affects functional parameters (such as motility, viability, acrosome membrane integrity and mitochondrial membrane potential) [9,17,18].

Horse sperm are particularly susceptible to oxidative stress due to the large amount of polyunsaturated fatty acids (PUFAs) in plasma membrane phospholipids [13,19,20], and their low intracellular antioxidant activity [21,22,23,24]. Under this scenario, seminal plasma (SP) represents the main antioxidant defense source, as it is endowed with enzymatic and non-enzymatic antioxidants that scavenge excessive ROS [25,26,27,28]. Yet, current equine semen cryopreservation protocols remove SP by centrifugation [14,29]. While this step is carried out to improve post-thaw sperm quality [1,30], it entails the removal of antioxidants, thus increasing further the susceptibility of sperm to oxidative damage [24,31].

In the horse, as in other mammalian species, differences in sperm cryotolerance between breeds [32], individuals [33] and even between ejaculates from the same animal [6] have been reported. Because these differences, which allow ejaculates to be classified as with “good” (GFE) or “poor” freezability (PFE) [14,28,34], are related to the composition of SP [35,36], elucidating the function of antioxidants is very relevant. In horses, some SP antioxidant enzymes (such as superoxide dismutase (SOD), catalase (CAT) and the glutathione peroxidase/glutathione reductase system (GPX/GSR)) are responsible for reducing oxidative stress [14,26,37]. Of these enzymes, total and specific activities of SOD in horse SP are related to sperm cryotolerance [14]. Moreover, paraoxonase type 1 (PON1) is another antioxidant enzyme that has been found in the SP of humans [38,39], pigs [40] and donkeys [28]. This enzyme could also play a function, as its activity levels are known to be positively associated to several sperm quality and functionality parameters in liquid-stored pig semen [41], and frozen-thawed pig [42] and donkey [28] sperm. Regarding the non-enzymatic antioxidant capacity of SP, it has been analyzed in other mammalian species using different methodological approaches, such as cupric-reducing antioxidant capacity (CUPRAC), ferric-reducing ability of plasma (FRAP) and Trolox equivalent antioxidant capacity (TEAC). Recently, these non-enzymatic antioxidant capacities were found to be directly involved in the resilience of pig [42] and donkey [28] sperm to freeze-thawing.

A balance between ROS production and antioxidants is crucial for sperm function and survival [9,43]. In this sense, it has been established that the measurement of the oxidative stress index (OSI) accurately shows the oxidant/antioxidant ratio in a biological sample; hence, an increase in this ratio would indicate the risk of oxidative stress due to an increase in ROS production or a decrease in antioxidants [44]. This new concept has been applied to SP in other species, such as pigs [45] and donkeys [28]. In both cases, the levels of SP-OSI appear to be related to the sperms’ ability to withstand liquid storage in pigs [45] and cryopreservation in donkeys [28].

Against this background, the present study aimed to evaluate whether OSI, activities of enzymatic antioxidants (including PON1, SOD, CAT, and GPX) and capacity of non-enzymatic antioxidants (measured in terms of: CUPRAC, FRAP, and TEAC) are related to horse sperm cryotolerance.

## 2. Materials and Methods

### 2.1. Stallions and Samples

Ejaculates were collected from 21 stallions (one ejaculate per individual) of different breeds, mature (between 5 and 15 years old) and of proven fertility. All stallions were clinically healthy and their standard diet included mixed hay and basic concentrate, without antioxidant supplementation and with *ad libitum* water availability. They were housed in individual paddocks at the facilities of the Equine Reproduction Service, Autonomous University of Barcelona (Bellaterra, Cerdanyola del Vallès, Spain). This is a center, approved (authorization number: ES09RS01E) by the European Union for the collection of equine semen, that operates under rigorous health and animal welfare protocols. This Service already works under the approval of the Regional Government of Catalonia, Spain; given that no manipulation on the animals beyond the collection of semen was conducted, the Ethics Committee of our institution indicated that no further ethical approval to carry out this study was necessary. Likewise, the health guidelines established by the Council of the European Communities in Directive 82/894/CEE of 21 December 1982 were complied with, as stallions were free from equine viral arteritis, equine infectious anemia and equine contagious metritis.

The collection was performed using a Hannover model artificial vagina (Minitüb GmbH, Tiefenbach, Germany), previously heated to a temperature between 48 °C and 50 °C and coupled with an in-line nylon filter to remove the gel fraction. Once the ejaculate was obtained, the gel fraction was removed and 10 µL were taken to evaluate the sperm concentration using a Neubauer chamber (Paul Marienfeld GmbH & Co. KG, Lauda-Königshofen, Germany). After that, each ejaculate was divided into two aliquots. The first was used to harvest the SP (see Section 2.2 for more details), and the second was diluted 1:5 (*v*:*v*) in a commercial extender based on skimmed milk [46] and subsequently cryopreserved (see Section 2.5 for more details). Before cryopreservation, this second aliquot was used for sperm quality analysis (which included sperm motility, morphology and viability). Sperm motility was analyzed using a Computer-Aided Sperm Analysis (CASA) system (see Section 2.6 for more details), and sperm viability and morphology through an eosin-nigrosine staining [47]. All semen samples used in this study fulfilled the standard thresholds for sperm quality before freezing (˃80% total motile sperm, ˃60% viable sperm, and ˃70% morphologically normal sperm).

### 2.2. Seminal Plasma (SP) Collection

Right after collection, ejaculates were centrifuged five times at 1500× *g* and 4 °C for 10 min (Medifriger BL-S; JP Selecta S.A., Barcelona, Spain). The supernatant was examined under a phase contrast microscope (Olympus Europe; Hamburg, Germany) at 200× to verify the absence of sperm. Finally, 5 mL of each SP sample was stored at −80 °C until analysis. Once all SP samples were procured and stored, they were thawed on ice to measure the activity levels of enzymatic and non-enzymatic antioxidants at the same time.

### 2.3. Measurement of Enzymatic and Non-Enzymatic Antioxidant Levels in SP

Enzymatic antioxidants assessed in SP were PON1, SOD, CAT, and GPX. Activity levels of PON1 were measured following the protocol described by Barranco et al. [40] adapted to horse SP, by measuring the hydrolysis of p-nitrophenyl acetate into p-nitrophenol. For the measurement of SOD, CAT, and GPX activity, commercially available assays were used following the manufacturer’s instructions (CAT: Merck, Darmstadt, Germany; GPX and SOD: Randox, Crumlin, UK). Determination of PON1, SOD, and GPX activity was conducted using an Olympus AU400 automated chemistry analyzer (Olympus Europe GmbH, Hamburg, Germany), whereas that of CAT was performed using a microplate reader (PowerWave XS; Bio-Tek Instruments, Winooski, VT, USA). Activity levels of PON1 and GPX were expressed as IU/L, whereas those of SOD and CAT were expressed as IU/mL.

Non-enzymatic antioxidant capacity of SP was measured in terms of CUPRAC, FRAP, and TEAC. All of them were assessed following the protocol described by Li et al. [42], adapted to horse SP. These assays are based on the reduction of Cu^2+^ to Cu^+^ (CUPRAC) [48] and of Fe^3+^ to Fe^2+^ (FRAP) [49], as well as a color change by 2,2′-azinobis-3-ethylbenzothiazoline-6-sulfonate (TEAC) [10]. All these determinations were performed using an Olympus AU400 automated chemistry analyzer. Activity levels of CUPRAC, FRAP, and TEAC were expressed as mmol Trolox equivalent/L.

The analytes from each SP sample were measured in duplicate. In each test, the intra and inter-assay coefficient showed a variation below 10%.

### 2.4. Measurement of Oxidative Stress Index (OSI) in SP

The levels of SP-OSI were calculated as follows: OSI (arbitrary unit) = Total oxidative status (TOS, μmol H_2_O_2_ equivalent/L)/TEAC (mmol Trolox equivalent/L) [28,50]. TOS was measured following the protocol described by Erel [51], adapted to horse SP. This assay is based on the oxidation of the ferrous ion to the ferric ion in the presence of oxidants in an acid medium, and the measurement of the ferric ion by xylenol orange. An Olympus AU400 automated chemistry analyzer was used for this evaluation. TOS results were expressed as μmol H_2_O_2_ equivalent/L.

### 2.5. Sperm Cryopreservation

Prior to cryopreservation, each extended semen sample was centrifuged at 660× *g* and 20 °C for 15 min. Thereafter, the supernatant was discarded and the pellet was resuspended in a commercial freezing extender (BotuCRIO^®^; Botupharma Animal Biotechnology, Botucatu, Brazil) containing 1% glycerol and 4% methylformamide as permeable cryoprotectants. After analyzing the sperm concentration and viability in each sample, the same freezing medium was added to obtain a final concentration of 200 × 10^6^ viable spermatozoa/mL. Finally, samples loaded into 0.5-mL straws were cooled/frozen using an automatic controlled-rate freezer (Ice-Cube 14S; Minitüb GmbH, Tiefenbach, Germany); the freezing curve included the following three stages: (1) cooling of 20 °C to 5 °C for 60 min at a rate of −0.25 °C/min, (2) freezing of 5 °C to −90 °C for 20 min at a rate of −4.75 °C/min, and (3) freezing from −90 °C to −120 °C for 2.7 min at a rate of −11.11 °C/min. Once this process was completed, straws were plunged into liquid nitrogen at −196 °C and stored in appropriate tanks for conservation. Thawing was carried out in a circulating water bath at 37 °C for 30 s; the content of each straw was poured into a 10-mL conical tube and further diluted (1:2, *v*/*v*) with Kenney extender [46], pre-warmed at 37 °C. Sperm quality and functionality parameters assessed in each frozen-thawed sample were: (1) motility, (2) plasma membrane and (3) acrosome integrity, (4) membrane lipid disorder, (5) mitochondrial membrane potential (MMP), and intracellular levels of (6) overall ROS and (7) superoxides (see Section 2.6 and Section 2.7 for more details).

### 2.6. Sperm Motility Analysis

Sperm motility analysis was performed using the CASA-Mot module of the ISAS^®^ v1.2 system (Proiser R + D, Valencia, Spain). This system consists of a high-resolution digital camera (model MQ003MG-CM; Proiser R + D) that captures up to 100 frames per second (fps). A reusable Spermtrack^®^10 chamber (Spk 10; Proiser R + D) pre-warmed to 37 °C was used, in which 2 µL of each sample was placed. Using a 10× negative phase contrast microscope (model UOP200i; Proiser R + D), a minimum of 500 spermatozoa were counted per analysis. The analysis included total (TM, %) and progressive (PM, %) motility, together with the kinematic parameters that define sperm movement: straight line velocity (VSL, µm/s), curvilinear velocity (VCL, µm/s), average path velocity (VAP, µm/s), straightness coefficient (STR = [VSL/VAP] × 100, %), linearity coefficient (LIN = [VSL/VCL] × 100, %), wobble coefficient (WOB = [VAP/VCL] × 100, %), beat-cross frequency (BCF, Hz), and amplitude of lateral head displacement (ALH, µm). In all the analyses, the CASA-Mot settings recommended by the manufacturer were used (particle area > 4 and < 75 µm^2^, connectivity = 6, minimum number of images to calculate ALH = 10) and cut-off values were taken for total (VAP ≥ 10 µm/s) and progressive (STR ≥ 75%) motility. Three replicates per sample were examined.

### 2.7. Sperm Functionality Analysis

Sperm functionality parameters were analyzed by flow cytometry and included: plasma membrane integrity (SYBR14/PI), acrosome integrity (*Arachis hypogaea* (peanut) agglutinin-fluorescein isothiocyanate (PNA-FITC)/Propidium iodide (PI)), mitochondrial membrane potential (MMP; 5,5′,6,6′-tetrachloro-1,1′3,3′tetraethyl-benzimidazolylcarbocyanine iodide (JC-1)), intracellular levels of ROS (2,7-dichlorodihydrofluorescein and diacetate (H_2_DCFDA)/PI) and superoxides (hydroethidine (HE)/1-(4-[3-methyl-2,3-dihydro-(benzo-1,3-oxazole)-2-methylidene]-quinolinium)-3-trimethylammonium propane diodide (YO-PRO-1)), and plasma membrane lipid disorder (Merocyanine 540 (M540)/YO-PRO-1). The flow cytometer used was a CytoFLEX (Beckman Coulter Fullerton, CA, USA) with a sheath flow rate set at 10 µL/min. Fluorochromes were purchased from Molecular Probes^®^ (Thermo Fisher Scientific, Waltham, MA, USA) and were resuspended in dimethyl sulfoxide (DMSO; Merck). Analyses were performed following the recommendations of the International Society for Advance Cytometry (ISAC) [52]. Prior to staining, sperm concentration was adjusted to 1 × 10^6^ sperm/mL. In each sample, a total of 10,000 events were analyzed and three technical replicates were evaluated.

Samples were excited with an argon ion laser (488 nm) at a power of 50 mW. Distributions of two different dot plots were used to exclude (1) cellular aggregates, based on the dot plot distribution of forward scatter height (FSC-H) and altitude (FSC-A), and (2) cellular debris, as a function of FSC-A distribution and side scatter altitude (SSC-A) dot plots. Four different optical filters were used: (1) FITC with a band-pass of 525-540 nm for analysis of SYBR14, PNA-FITC, JC-1 monomers (JC-1_mon_), dichlorofluorescein (DCF^+^) and (YO-PRO-1), (2) PE with a band-pass of 585-542 nm for analysis of JC-1 aggregates (JC-1_agg_) and fluorescent ethidium (E^+^), (3) ECD with a band-pass of 610–620 nm for analysis of M540, and (4) PC5.5 with a band-pass of 690–650 nm for analysis of PI. The information obtained regarding each event (FSC-A, FSC-H, SSC-A, FITC, PE and PC5.5) was collected in xit files and analyzed using the CytExpert analysis software (Beckman Coulter Fullerton) to quantify sperm populations. For each parameter, the corresponding mean and standard error of the mean (SEM) were calculated.

#### 2.7.1. Plasma Membrane Integrity (SYBR14/PI)

Plasma membrane integrity of sperm was analyzed using the LIVE/DEAD sperm viability kit (SYBR14/PI), according to the protocol described by Garner and Johnson [53], adapted to horse sperm. Briefly, semen samples were incubated with SYBR14 (final concentration: 31.8 nM) for 10 min and then with PI (final concentration: 7.6 µM) for 5 min in the dark at 37 °C. Three sperm populations were identified: (1) sperm with a damaged plasma membrane (SYBR14^+^/PI^+^), (2) sperm with a damaged plasma membrane (SYBR14^−^/PI^+^), and (3) sperm with an intact plasma membrane (SYBR14^+^/PI^−^; viable sperm). Particles without staining (SYBR14^−^/PI^−^) were considered as non-sperm debris and were used to correct the data in the other assessments. SYBR14 overflow in channel PC5.5 (8.34%) was compensated.

#### 2.7.2. Acrosome Integrity (PNA-FITC/PI)

Acrosome integrity of sperm was analyzed using the combination of PNA-FITC and PI, according to the protocol described by Rathi et al. [54]. Briefly, semen samples were incubated in the dark at 37 °C for 10 min with PNA conjugated with FITC (final concentration: 1.17 µg/mL) and with PI (final concentration: 5.6 µM). Four sperm populations were identified: (1) sperm with damaged plasma membrane (PNA-FITC^+^/PI^−^), (2) sperm with damaged plasma membrane together with fully-lost outer acrosome membrane (PNA-FITC^−^/PI^+^), (3) sperm with a damaged plasma membrane that exhibited an outer acrosome membrane that could not be completely intact (PNA-FITC^+^/PI^+^), and (4) viable sperm with an intact acrosome membrane (PNA-FITC^−^/PI^−^). No compensation was needed.

#### 2.7.3. Mitochondrial Membrane Potential (JC-1)

Mitochondrial membrane potential (MMP) of sperm was analyzed using JC-1, according to the protocol described by Ortega-Ferrusola et al. [55]. In brief, semen samples were incubated with JC-1 (final concentration: 750 nM) at 37 °C for 30 min in the dark. JC-1 molecules form orange fluorescent aggregates (JC-1_agg_) in the presence of high MMP, while remaining as green fluorescent monomers (JC-1_mon_) in the presence of low MMP. Two sperm populations were identified: (1) sperm exhibiting low MMP (JC-1_mon_ fluorescence intensity higher than JC-1_agg_), and (2) sperm exhibiting high MMP (JC-1_agg_ fluorescence intensity higher than JC-1_mon_). In each population, the fluorescence intensity of JC-1_mon_ and JC-1_agg_ was recorded and the ratio between them was calculated. Data were not compensated.

#### 2.7.4. Intracellular Reactive Oxygen Species: Total ROS (H_2_DCFDA/PI) and O_2_^−^ (HE/YO-PRO-1)

Intracellular ROS levels of sperm were analyzed using oxidation-sensitive fluorescent probes: H_2_DCFDA to assess overall ROS and HE to measure superoxide anion (O_2_^−^) [56]. The differentiation of viable and non-viable spermatozoa was performed using PI (for H_2_DCFDA) or YO-PRO-1 (for HE), following the modified protocol of Guthrie and Welch [57].

For overall ROS measurement, semen samples were incubated with H_2_DCFDA (final concentration: 50 µM) at 37 °C in the dark for 20 min, and then with PI (final concentration: 6 µM) for 5 min. In presence of ROS, there is a de-esterification and oxidation of H_2_DCFDA to DCF^+^, which is a highly fluorescent molecule. Four sperm populations were identified: (1) non-viable sperm with low ROS levels (DCF^−^/PI^+^), (2) viable sperm with low ROS levels (DCF^−^/PI^−^), (3) non-viable sperm with high ROS levels (DCF^+^/PI^+^), and (4) viable sperm with high ROS levels (DCF^+^/PI^−^). DCF^+^ fluorescence intensity was recorded in all sperm populations. Data were not compensated.

For O_2_^−^ analysis, semen samples were incubated with HE (final concentration: 5 µM) and YO-PRO-1 (final concentration: 31.25 nM) at 37 °C in the dark for 30 min. In the presence of O_2_^-^, HE is oxidized into E^+^. Four sperm populations were identified: (1) non-viable sperm with low O_2_^−^ levels (E^−^/YO-PRO-1^+^), (2) viable sperm with low O_2_^−^ levels (E^−^/YO-PRO-1^−^), (3) non-viable sperm with high O_2_^−^ levels (E^+^/YO-PRO-1^+^), and (4) viable sperm with high O_2_^−^ levels (E^+^/YO-PRO-1^−^). Fluorescence intensity of E^+^ was recorded in all sperm populations. The overflow of E^+^ to the FITC channel (3.62%) was compensated.

#### 2.7.5. Plasma Membrane Lipid Disorder (M540/YO-PRO-1)

Plasma membrane lipid disorder of sperm was analyzed using the combination of M540/YO-PRO-1, according to the protocol described by Rathi et al. [54] with minor modifications [58], adapted to horse sperm. Briefly, semen samples were incubated in the dark at 37 °C for 10 min with M540 (final concentration: 2.5 µM) and YO-PRO-1 (final concentration: 25 nM). Four sperm populations were identified: (1) viable sperm with high plasma membrane lipid disorder (M540^+^/YO-PRO-1^−^), (2) non-viable sperm with high plasma membrane lipid disorder (M540^+^/YO-PRO-1^+^), (3) viable sperm with low plasma membrane lipid disorder (M540^−^/YO-PRO-1^−^), and (4) non-viable sperm with low plasma membrane lipid disorder (M540^−^/YO-PRO-1^+^). Data were not compensated.

### 2.8. Statistical Analysis

The analysis of the data and the elaboration of the figures was carried out using the statistical package R (V 4.0.3, R Core Team; Vienna, Austria) and the GraphPad Prism software (V 8.4.0, GraphPad Software LLC; San Diego, CA, USA), respectively. The first step was to check the normality of data using the Shapiro–Wilk test, as well as the homogeneity of variances using the Levene test. When necessary, arcsine √x was applied to transform data and thus obtain a normal distribution. In all cases, the minimum level of statistical significance was set at *p* ≤ 0.05.

#### 2.8.1. Classification of Ejaculates Based on Their Cryotolerance

Classification of the 21 horse ejaculates based on their cryotolerance (GFE and PFE) was performed following the procedure described by Morató et al. [34]. In brief, post-thaw percentages of TM and of sperm with an intact plasma membrane (sperm viability, SYBR14^+^/PI^−^) recorded in each sample were taken to perform a complete linkage hierarchical cluster analysis using Euclidean distances from the mentioned parameters. Results in the text are expressed as means ± SEM.

#### 2.8.2. Comparison of Antioxidant Variables Measured in SP between Good (GFE) and Poor Freezability Ejaculates (PFE) 

Comparison between GFE and PFE regarding activity levels of enzymatic (PON1, SOD, CAT, and GPX) and non-enzymatic (CUPRAC, FRAP, and TEAC) antioxidants in SP, and SP-OSI was performed using a *t*-test for independent samples. When, even after transforming the data, a normal distribution was not obtained, the Mann–Whitney test was used as a non-parametric alternative. Results in the text were expressed as means ± SEM.

#### 2.8.3. Correlations of Antioxidant Variables with Post-Thaw Semen Parameters

The relationship of activity levels of enzymatic antioxidants (PON1, SOD, CAT, and GPX) and non-enzymatic antioxidants (CUPRAC, FRAP, and TEAC) in SP, and SP-OSI with motility parameters (TM, PM, VCL, VSL, VAP, LIN, STR, WOB, ALH, and BCF) and with sperm functionality parameters (SYBR14^+^/PI^−^, PNA-FITC^−^/PI^−^, JC-1_agg_, DCF^+^/PI^−^, E^+^/YO-PRO-1^−^, and M540^+^/YO-PRO-1^−^) was evaluated by calculating Pearson’s correlation coefficients.

## 3. Results

### 3.1. Classification of Horse Ejaculates Based on Their Cryotolerance

Classifying the 21 horse ejaculates through hierarchical clusters (*p* < 0.001) based on post-thaw TM and sperm viability (SYBR14^+^/PI^−^) separated 13 GFE from 8 PFE. Figure 1 shows that the ejaculates classified as GFE exhibited significantly higher values (*p* < 0.001) of both TM and sperm viability (SYBR14^+^/PI^−^) compared to those classified as PFE (66.44 ± 2.61% vs. 35.54 ± 4.36%, and 67.90 ± 1.54% vs. 42.03 ± 2.49%, respectively). The ranges of TM and sperm viability (SYBR14^+^/PI^−^) in both groups of ejaculates are shown in Appendix A, respectively.

### 3.2. Activity Levels of Enzymatic Antioxidants in the SP of Horse Ejaculates Classified as GFE and PFE

The ejaculates classified as GFE showed significantly higher activity levels (*p* < 0.05) of PON1 and SOD in their SP than those classified as PFE (0.80 ± 0.08 IU/L vs. 0.58 ± 0.06 IU/L, and 245.23 ± 35.52 IU/mL vs. 171.38 ± 24.19 IU/mL, respectively, Figure 2a,b). No differences regarding the activity levels of CAT and GPX in SP (Figure 2c,d) were found between GFE and PFE. The mean ± SEM and ranges of the activity levels of enzymatic antioxidants measured in the SP of all horse ejaculates are shown in Appendix A.

### 3.3. Activity Levels of Non-Enzymatic Antioxidants in the SP of Horse Ejaculates Classified as GFE and PFE

The SP-levels of non-enzymatic antioxidants measured in terms of CUPRAC and FRAP did not differ between GFE and PFE (Figure 3a,b). The levels of TEAC in SP, however, were significantly higher (*p* < 0.001) in ejaculates classified as GFE than in those classified as PFE (0.83 ± 0.06 mmol/L vs. 0.47 ± 0.05 mmol/L, respectively; Figure 3c). The mean ± SEM and ranges of non-enzymatic antioxidants measured in the SP of all ejaculates are shown in Appendix A.

### 3.4. Oxidative Stress Index in the SP of Horse Ejaculates Classified as GFE and PFE

Ejaculates classified as GFE showed significantly lower SP-OSI levels (*p* < 0.001) compared to those classified as PFE (6.35 ± 0.48 vs. 13.03 ± 1.80; Figure 4). The SP levels of TOS in ejaculates classified as GFE were 4.99 ± 0.18 µmol/L, and in those classified as PFE were 5.53 ± 0.31 µmol/L (*p* > 0.05). The mean ± SEM and ranges of levels OSI and TOS measured in the SP of all horse ejaculates are shown in Appendix A.

### 3.5. Correlations between SP Antioxidants (Enzymatic, Non-Enzymatic and OSI) and Post-Thaw Sperm Motility Parameters

Figure 5 shows the correlations of enzymatic and non-enzymatic antioxidants, and OSI levels measured in the SP of horse ejaculates with post-thaw sperm motility parameters. Activity levels of four enzymatic antioxidants (PON1, SOD, CAT, and GPX) were positively correlated (*p* < 0.05) with PM (PON1: r = 0.46; SOD: r = 0.47; CAT: r = 0.53; GPX: r = 0.48). In addition, activity levels of PON1 and CAT were also positively correlated with TM (r = 0.50, *p* < 0.05; and r = 0.61, *p* < 0.01, respectively) and with several kinematic parameters such as: VCL (r = 0.44, *p* < 0.05; and r = 0.56, *p* < 0.01, respectively), VAP (r = 0.50, *p* < 0.05; and r = 0.51, *p* < 0.05, respectively), VSL in the case of PON1 (r = 0.45, *p* < 0.05), and ALH in the case of CAT (r = 0.53, *p* < 0.05). For non-enzymatic antioxidants, only TEAC was positively correlated with TM (r = 0.65, *p* < 0.01) and PM (r = 0.62, *p* < 0.01), as well as with the following kinematic parameters: VCL (r = 0.62, *p* < 0.01), VSL (r = 0.51, *p* < 0.05), VAP (r = 0.54, *p* < 0.05), and ALH (r = 0.62, *p* < 0.01). Finally, OSI was negatively correlated with TM (r = −0.72, *p* < 0.001) and PM (r = −0.65, *p* < 0.01), and the following kinematic parameters: VCL (r = − 0.68, *p* < 0.001), VSL (r = −0.60, *p* < 0.01), VAP (r = −0.64, *p* < 0.01), ALH (r = −0.67, *p* < 0.001), and BCF (r = −0.43, *p* < 0.05). The mean ± SEM and ranges of each sperm motility parameter in GFE and PFE recorded post-thaw are shown in Appendix A.

### 3.6. Correlations between SP Antioxidants (Enzymatic, Non-Enzymatic and OSI) Levels and Post-Thaw Sperm Functionality Parameters

Figure 6 shows the correlations of enzymatic and non-enzymatic antioxidants, and OSI measured in the SP of horse ejaculates with post-thaw sperm functionality parameters. Enzymatic antioxidant activity levels were positively correlated with several sperm functionality parameters: PON1 and CAT with plasma membrane integrity (SYBR14^+^/PI^−^; r = 0.46, *p* < 0.05; r = 0.45, *p* < 0.05, respectively), SOD and GPX with acrosome and plasma membrane integrity (PNA-FITC^−^/PI^−^; r = 0.59, *p* < 0.01; and r = 0.67, *p* < 0.001, respectively), and GPX with the proportion of viable sperm with high MMP (JC-1_agg_; r = 0.46, *p* < 0.05). Similarly, non-enzymatic antioxidants were positively correlated with several sperm functionality parameters: TEAC with plasma membrane integrity (SYBR14^+^/PI^−^; r = 0.57, *p* < 0.01), and CUPRAC and TEAC with acrosome and plasma membrane integrity (PNA-FITC^−^/PI^−^; r = 0.52, *p* < 0.05; and r = 0.46, *p* < 0.05, respectively). Finally, OSI was negatively correlated with plasma membrane integrity (SYBR14^+^/PI^−^; r = −0.70, *p* < 0.001) and with acrosome and plasma membrane integrity (PNA-FITC^−^/PI^−^; r = −0.46, *p* < 0.05), and positively with the proportion of viable sperm with high plasma membrane lipid disorder (M540^+^/YO-PRO-1^−^; r = 0.41, *p* < 0.05). The mean ± SEM and ranges of each sperm functionality parameter in GFE and PFE recorded post-thaw are shown in Appendix A.

### 3.7. Correlations between Seminal Plasma Antioxidants (Enzymatic, Non-Enzymatic and OSI) in Horse Ejaculates

Figure 7 shows the correlations between OSI, enzymatic and non-enzymatic antioxidants measured in the SP of horse ejaculates. Levels of enzymatic and non-enzymatic antioxidants showed positive correlations with each other: PON1 and CAT with GPX (r = 0.57, *p* < 0.01; and r = 0.48, *p* < 0.05, respectively) and with TEAC (r = 0.51, *p* < 0.05, and r = 0.50, *p* < 0.05, respectively); SOD with GPX (r = 0.86, *p* < 0.001) and with non-enzymatic antioxidants (CUPRAC: r = 0.71, *p* < 0.001; FRAP: r = 0.62, *p* < 0.01, and TEAC: r = 0.75, *p* < 0.001); GPX with non-enzymatic antioxidants (CUPRAC: r = 0.72, *p* < 0.001; FRAP: r = 0.64, *p* < 0.01, and TEAC: r = 0.75, *p* < 0.001); CUPRAC with FRAP (r = 0.92, *p* < 0.001) and with TEAC (r = 0.62, *p* < 0.01); and FRAP with TEAC (r = 0.54, *p* < 0.05). On the other hand, OSI was negatively correlated with PON1 (r = −0.49, *p* < 0.05), SOD (r = −0.49, *p* < 0.05), GPX (r = −0.54, *p* < 0.05), and with TEAC (r = −0.85, *p* < 0.001). The mean ± SEM and ranges of each enzymatic and non-enzymatic antioxidant’s levels, as well as OSI measured in the SP of all horse ejaculates, are shown in Appendix A.

## 4. Discussion

The present study showed that some SP-components, such as enzymatic (PON1 and SOD) and non-enzymatic (measured in terms of TEAC) antioxidants, and SP-OSI, are related to sperm cryotolerance in the horse.

Cryopreservation is a stressful process for the sperm, resulting in elevated ROS production [6]. While low levels of ROS have been reported to be beneficial for physiological sperm processes, such as sperm motility, capacitation, and acrosome reaction [59], excessive ROS lead to an imbalance in cellular antioxidant defense and, ultimately, to oxidative stress [26,60]. Due to their limited antioxidant capacity, sperm cannot counteract high levels of ROS [23,61], meaning that the antioxidant responsibility relies on SP, which is removed before freezing [14,29]. It seems, however, that the brief contact between sperm and SP before its removal could be enough for SP antioxidants to exert a beneficial effect upon sperm cryotolerance [14,27,28]. In this sense, the great variability in sperm cryotolerance between stallions could be related to the composition of their SP [14]. Our results agree with this hypothesis, as there were differences between enzymatic, non-enzymatic antioxidants and SP-OSI between GFE and PFE, and these SP components were even correlated to some post-thaw sperm quality and functionality parameters.

Regarding enzymatic antioxidants, we quantified, for the first time in horses, the activity levels of PON1 in SP and we evaluated its relationship with sperm cryotolerance. The results evidenced that the activity levels of this enzyme were higher in the SP of GFE than in that of PFE. These results agree with previous studies conducted in pigs [42] and, more recently, in donkeys [28]. It is also worth noting that the activity levels of PON1 in horse SP were higher (approximately more than two-fold) than those reported in pigs and donkeys, which could represent a species-specificity regarding the antioxidant function of this enzyme. In this regard, the enzyme could play a key role as an antioxidant in horse semen. This hypothesis would be supported by the positive relationship found between PON1, some sperm motility parameters and plasma membrane integrity. In fact, the negative effect of ROS on horse sperm motility is well-known, mainly due to the plasma membrane damage resulting from lipoperoxidation [9,17,18]. In addition, Aitken et al. [62] associated the loss of sperm motility with an increase in the availability of homocysteine thiolactone, a cyclic congener of homocysteine, in sperm. Homocysteine is a non-protein α-amino acid that differs from cysteine by having an additional methylene bridge; elevated levels of both homocysteine and homocysteine thiolactone may induce oxidative stress [63]. As a product of plasma membrane lipoperoxidation, lipid aldehydes, such as 4-hydroxynonenal (4-HNE) [64,65], are generated; this favors the accumulation of homocysteine thiolactone in sperm, thus inhibiting PON1 activity [62]. Blockade of PON1 facilitates the direct interaction of homocysteine thiolactone with the ε-amino group of lysine residues in sperm proteins, triggering a series of changes that end up with a reduction of sperm motility [12]. Bearing this in mind, it is reasonable to suggest that PON1 in SP may be an essential ROS scavenger in horse semen.

Similarly, our results confirmed that the SOD present in SP is relevant for the cryotolerance of horse sperm, because its activity levels were higher in GFE than in PFE. Furthermore, a positive relationship between SOD activity in SP and some post-thaw sperm quality parameters, such as progressive motility and acrosome membrane integrity, was observed. These results are consistent with those previously reported in horse [14], donkey [27,28], pig [42], buffalo [66], and human SP [67]. SOD is an essential antioxidant to avoid oxidative stress because it initiates a cascade of reactions intended to neutralize the most harmful ROS [42,68]. Specifically, SOD neutralizes O_2_^-^, which, in excess, triggers a dangerous oxidative chain reaction leading to lipoperoxidation of plasma membrane and thereby leads to the loss of motility and sperm death [15]. This would explain the positive relationship of SOD activity with sperm motility and acrosome integrity, as well as the differences between GFE and PFE observed herein and in previous studies with frozen-thawed horse [14] and donkey [27,28] sperm. In addition, it is important to mention that the activity levels of SOD in horse SP observed in this work were lower than those reported in a recent study in donkey SP [28], but higher than those observed in pigs [42], which would indicate a great inter-species variability in SP composition.

As far as CAT and GPX are concerned, we did not find differences in their activity levels between GFE and PFE. This would be in agreement with previous findings in horses [14] and donkeys [27]. In our study, nevertheless, a positive relationship between the activity levels of these antioxidant enzymes and some post-thaw sperm quality and functionality parameters was observed. Supplementing the freezing medium with CAT has a positive impact on post-thaw sperm quality in pigs [69], bulls [70], and humans [71]. Similarly, the addition of other antioxidant enzymes, such as GPX, to the freezing media increases the cryotolerance of buffalo [66], dog [72], and cattle sperm [73]. Dismutation of O_2_^−^ by SOD ends in the formation of hydrogen peroxide (H_2_O_2_) and molecular oxygen (O_2_); CAT detoxifies the excess of H_2_O_2_ and GPX maintains H_2_O_2_ balance [37,74]. The CAT present in horse SP, therefore, seems to play a key role in preventing the accumulation of H_2_O_2_, which is known to negatively affect sperm in this species [75]. This would be in agreement with our results, as CAT activity in SP was positively related to the percentage of sperm with intact plasma membrane. On the other hand, the H_2_O_2_ initially generated after the action of SOD enters the oxidative/reductive balance system mediated by GPX, which is able maintain this ROS within the physiological levels without altering sperm function and survival [37,74]. This would be in accordance with our results because a positive relationship between GPX activity and acrosome integrity was found. Moreover, the linkage of CAT with sperm motility parameters has been observed in the liquid-stored semen of other mammalian species [61], such as dogs [76] and sheep [77], and GPX activity has been found to be positively related to the quality of fresh horse semen [78], and frozen-thawed cattle [79] and donkey [28] semen. Curiously, in our study, a positive relationship between GPX activity levels and mitochondrial membrane potential was observed, which could help explain the relationship between GPX and sperm motility.

Regarding non-enzymatic antioxidants in horse SP (measured in terms of CUPRAC, FRAP, and TEAC), only TEAC had a significant relevance for sperm cryotolerance, showing higher levels in the SP of GFE than in that of PFE. These data are in agreement with previous research in frozen-thawed pig [42] and donkey sperm [28]. TEAC evaluates the antioxidant effect of low molecular weight molecules (such as uric acid, α-tocopherol, bilirubin and ascorbic acid (AA)), and of those that contain sulfhydryl groups (-SH) in their structure (such as albumin) [80,81]. The antioxidant molecules containing -SH exert protection from the oxidative damage caused by ROS during cryopreservation, because they act on disulfide bonds between chromatin fibers, thus maintaining the integrity of the nucleoprotein structure [82,83,84]. In this study, we also found that TEAC in SP was positively related to post-thaw sperm plasma membrane integrity, which was in agreement with previous observations in frozen-thawed donkey sperm [28]. In the present study, we also found that TEAC in SP was positively related to acrosome integrity of post-thaw sperm. Consequently, non-enzymatic antioxidants of SP (assessed by TEAC) could be key to maintaining the functional integrity of cryopreserved horse sperm, as the positive relationship between TEAC in SP and post-thaw sperm motility parameters demonstrated. This would be also in line with the results of the aforementioned studies in frozen-thawed pig [42] and donkey [28] sperm. It is also worth noting that the fact that the CUPRAC of SP did not differ between GFE and PFE was in agreement with previous results obtained in frozen-thawed pig sperm [42]. In contrast, SP-FRAP in donkey [28] and pig semen [42], and SP-CUPRAC in donkey semen [28] were found to differ between GFE and PFE. These inconsistent results between species could be explained by differences in the composition of their SP and their sperm cryotolerance [14,27,28,37,85], as well as by the contribution of individual antioxidants to FRAP (such as AA, α-tocopherol, and mainly uric acid) and CUPRAC (such as thiols (including GSH)) [86].

Maintaining a balance between ROS and antioxidant levels is essential for optimal sperm function [9]. While an excess of ROS production can lead to the generation of oxidative stress, controlled ROS levels are essential for sperm physiological processes [59]. In this sense, OSI analysis is considered an accurate method to measure the oxidant/antioxidant ratio in biological samples [44]. Our results reveled that SP-OSI values, which were lower in GFE than in PFE, were negatively related to post-thaw sperm motility, plasma membrane and acrosome membrane integrity. These findings demonstrate that an adequate balance between oxidants and antioxidants in SP is crucial for the ability of horse sperm to withstand freeze-thawing. Similar evidence was reported in liquid-stored pig semen [45] and recently in frozen-thawed donkey sperm [28]. It is well known that an excess of ROS has a negative impact on the activity of some essential enzymes for sperm motility, such as glucose-6-phosphate dehydrogenase [87]. In addition, a ROS/antioxidants imbalance may cause mitochondrial dysfunction, leading to adenosine triphosphate depletion; this decrease in the energy available to sperm results in a reduction of their motility [88]. Interestingly, OSI values in horse SP were higher than those reported in donkey [28] and in pig SP [45]. One may, therefore, posit that the control of oxidative stress is highly complex and species-specific [14]. This hypothesis would be supported by the lack of relationship between the different antioxidants (enzymatic and non-enzymatic), SP-OSI with intracellular ROS levels in post-thaw sperm observed in this study, which differs from that previously reported in donkey [28] and pig [42] semen. Differences in SP-composition, sperm metabolism and reproductive strategy between species could also help explain these distinct outcomes.

Finally, we observed significant correlations between SP-OSI, enzymatic and non-enzymatic antioxidants; this demonstrates, also in horse semen, the great complexity of the oxidative stress control and maintenance of redox balance [89].

## 5. Conclusions

In conclusion, this research reported the relevance of enzymatic and non-enzymatic SP antioxidants, and OSI for horse sperm cryotolerance. This study quantified, for the first time in horse SP, the levels of PON1, of non-enzymatic antioxidant capacity (in terms of: CUPRAC, FRAP, and TEAC) and of the OSI. Our results revealed that the levels of some enzymatic (PON1 and SOD) and non-enzymatic (TEAC) antioxidants were higher in the SP of horse ejaculates classified as GFE than in those graded as PFE. A positive relationship between these SP components and some post-thaw sperm quality parameters was also observed (motility and plasma membrane integrity: PON1 and TEAC; acrosome membrane integrity: SOD and TEAC). In contrast, OSI in horse SP, which was higher in PFE than in GFE, was negatively associated with some sperm motility parameters, and plasma membrane and acrosome integrity, and positively with the percentage of viable sperm with high membrane lipid disorder. These findings support how important an adequate balance between oxidants and antioxidants in SP is for the ability of horse sperm to withstand freeze-thawing. In sum, the present work suggests that PON1, SOD, TEAC, and OSI in SP could be used as putative sperm cryotolerance biomarkers in the horse. Measuring these enzymatic and non-enzymatic antioxidants could allow us (1) to select those ejaculates that are likely to better withstand cryopreservation and (2) to identify which semen samples may need antioxidant supplements for their storage.

## Figures and Tables

**Figure 1 antioxidants-11-01279-f001:**
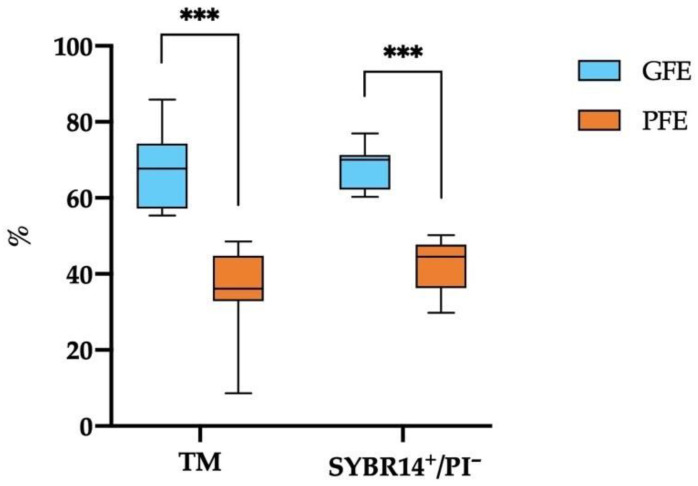
Box-whisker plot showing the percentages of total motile (TM) and viable (SYBR14^+^/PI^−^) sperm after thawing in horse ejaculates classified as with good (GFE, *n* = 13; blue) or poor freezability (PFE, *n* = 8; orange). The boxes enclose the 25th and 75th percentiles, the whiskers extend to the 5th and 95th percentiles, and the line indicates the median. (***) *p* ≤ 0.001.

**Figure 2 antioxidants-11-01279-f002:**
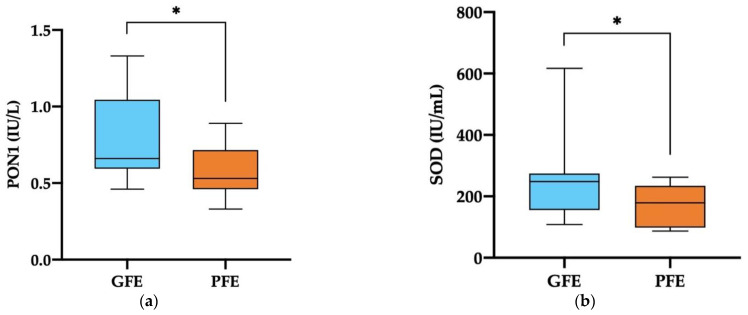
Box-whisker plot showing the activity levels (IU/mL) of enzymatic antioxidants measured in the seminal plasma of horse ejaculates classified as with good (GFE, *n* = 13; blue) or poor freezability (PFE, *n* = 8; orange): paraoxonase type 1 (PON1, **a**), superoxide dismutase (SOD, **b**), catalase (CAT, **c**), and glutathione peroxidase (GPX, **d**). The boxes enclose the 25th and 75th percentiles, the whiskers extend to the 5th and 95th percentiles, and the line indicates the median. (*) *p* ≤ 0.05.

**Figure 3 antioxidants-11-01279-f003:**
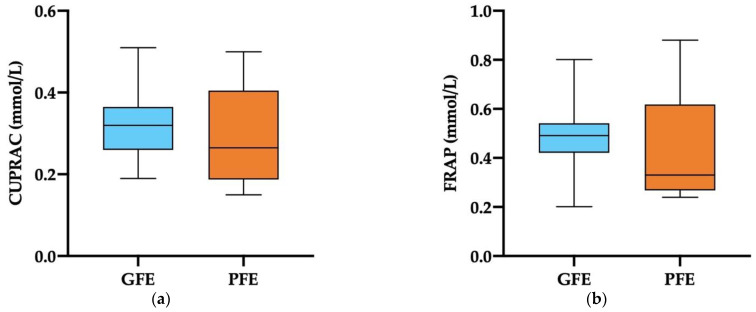
Box-whisker plot showing non-enzymatic antioxidants in the seminal plasma of horse ejaculates classified as with good (GFE, *n* = 13; blue) or poor freezability (PFE, *n* = 8; orange), measured as cupric-reducing antioxidant capacity (CUPRAC, **a**), ferric-reducing ability of plasma (FRAP, **b**), and Trolox equivalent antioxidant capacity (TEAC, **c**). The boxes enclose the 25th and 75th percentiles, the whiskers extend to the 5th and 95th percentiles, and the line indicates the median. (***) *p* ≤ 0.001.

**Figure 4 antioxidants-11-01279-f004:**
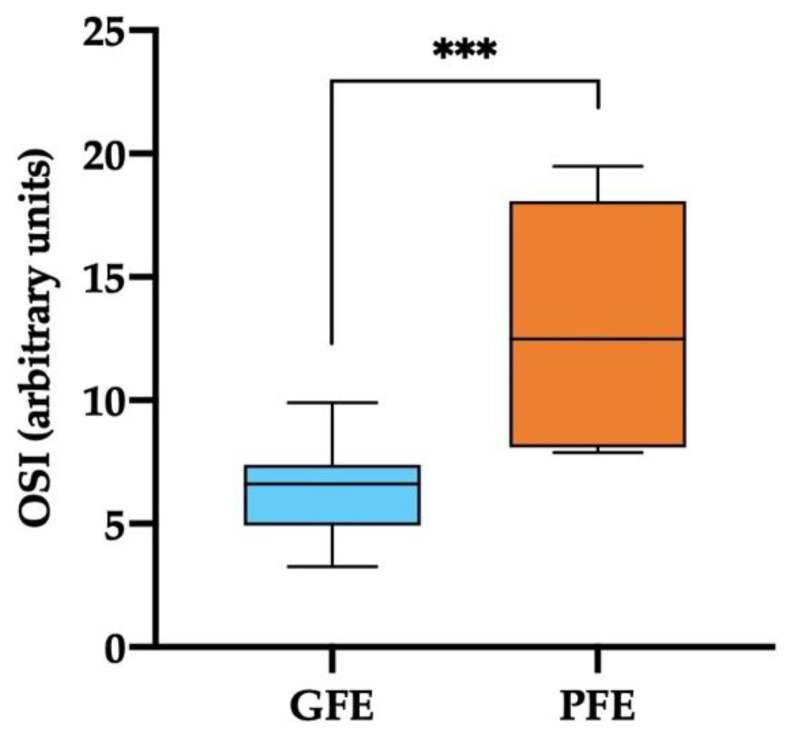
Box-whisker plot showing oxidative stress indexes (OSI) (arbitrary units) measured in the seminal plasma of horse ejaculates classified as with good (GFE, *n* = 13; blue) or poor freezability (PFE, *n* = 8; orange). The boxes enclose the 25th and 75th percentiles, the whiskers extend to the 5th and 95th percentiles, and the line indicates the median. (***) *p* ≤ 0.001.

**Figure 5 antioxidants-11-01279-f005:**
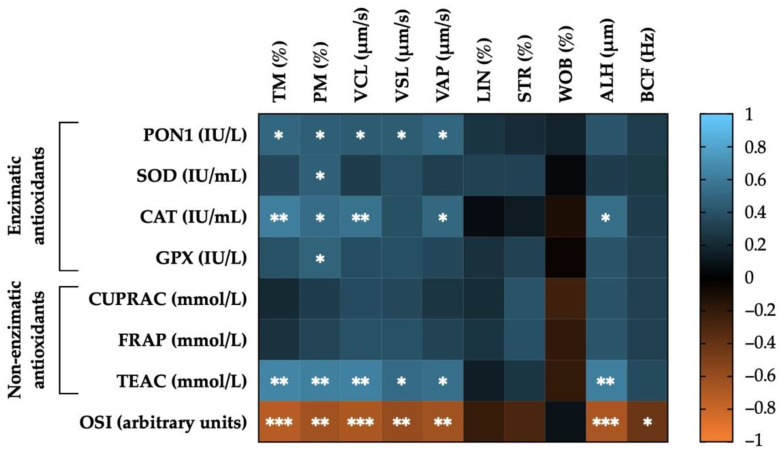
Heat map showing the correlations of enzymatic (paraoxonase type 1, PON1; superoxide dismutase, SOD; catalase, CAT; and glutathione peroxidase, GPX) and non-enzymatic antioxidants (measured in terms of: cupric-reducing antioxidant capacity, CUPRAC; ferric-reducing ability of plasma, FRAP; and Trolox equivalent antioxidant capacity, TEAC), as well as of oxidative stress index (OSI) in the seminal plasma of horse ejaculates (*n* = 21) with sperm motility parameters recorded post-thaw (total motility, TM; progressive motility, PM; curvilinear velocity, VCL; straight line velocity, VSL; average path velocity, VAP; linearity coefficient, LIN; straightness coefficient, STR; wobble coefficient, WOB; amplitude of lateral head displacement, ALH; and beat-cross frequency, BCF). The colors on the scale (1 to −1) indicate whether the correlation is positive (blue) or negative (orange). (*) *p* ≤ 0.05; (**) *p* ≤ 0.01; (***) *p* ≤ 0.001.

**Figure 6 antioxidants-11-01279-f006:**
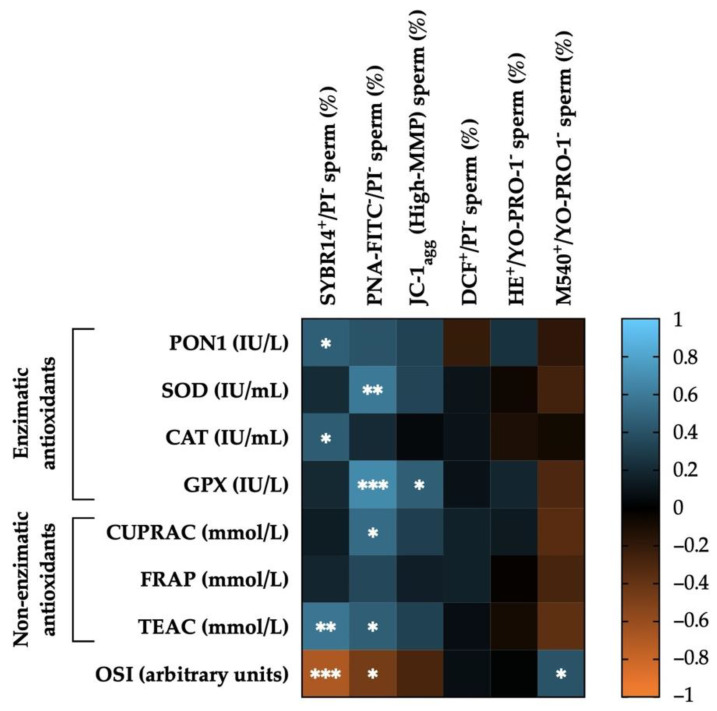
Heat map showing the correlations of enzymatic (paraoxonase type 1, PON1; superoxide dismutase, SOD; catalase, CAT; and glutathione peroxidase, GPX) and non-enzymatic antioxidants (measured in terms of: cupric-reducing antioxidant capacity, CUPRAC; ferric-reducing ability of plasma, FRAP; and Trolox equivalent antioxidant capacity, TEAC), as well as of oxidative stress index (OSI) in the seminal plasma of horse ejaculates (*n* = 21) with sperm functionality parameters recorded post-thaw (plasma membrane integrity, SYBR14^+^/PI^−^; acrosome membrane integrity, PNA-FITC^−^/PI^−^; mitochondrial membrane potential, MMP, JC-1_agg_; intracellular ROS levels, DCF^+^/PI^−^; intracellular superoxide levels, E^+^/YO-PRO-1^−^; and plasma membrane lipid disorder, M540^+^/YO-PRO-1^−^). The colors on the scale (1 to −1) indicate whether the correlation is positive (blue) or negative (orange). (*) *p* ≤ 0.05; (**) *p* ≤ 0.01; (***) *p* ≤ 0.001.

**Figure 7 antioxidants-11-01279-f007:**
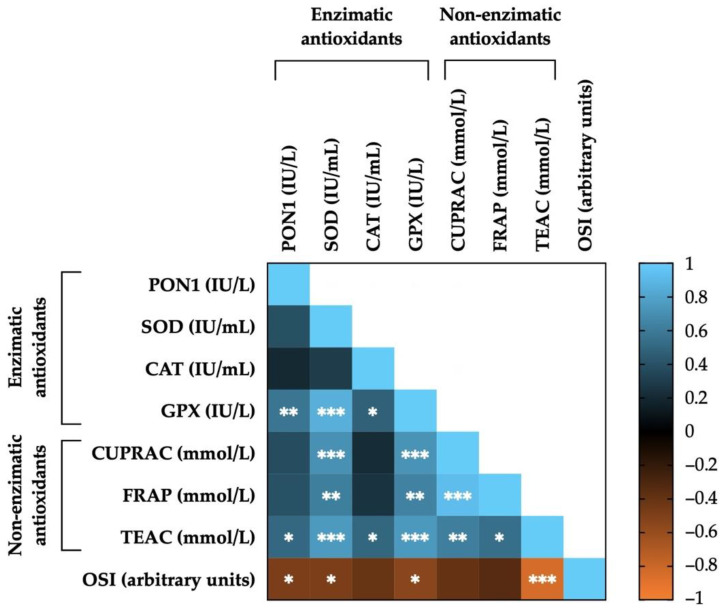
Heat map showing the correlations between enzymatic (paraoxonase type 1, PON1; superoxide dismutase, SOD; catalase, CAT; and glutathione peroxidase, GPX) and non-enzymatic antioxidants (measured in terms of: cupric-reducing antioxidant capacity, CUPRAC; ferric-reducing ability of plasma, FRAP; and Trolox equivalent antioxidant capacity, TEAC), as well as of oxidative stress index (OSI) measured in the seminal plasma of horse ejaculates (*n* = 21). The colors on the scale (1 to −1) indicate whether the correlation is positive (blue) or negative (orange). (*) *p* ≤ 0.05; (**) *p* ≤ 0.01; (***) *p* ≤ 0.001.

## Data Availability

All data is contained within the article and Appendix A.

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
