# Peer review of "Seminal Plasma Antioxidants Are Related to Sperm Cryotolerance in the Horse"

_antioxidants, 2022, doi:10.3390/antiox11071279_

Round 1

Reviewer 1 Report

1) It is important to test if addition of antioxidants to PFE would improve the parameters after freeing-thowing to the level of GFE? Moreover, how these parameters in GFE is affected by removing the antioxidants from the seminal plasma?

2) How extracellular enzymatic antioxidants affect intracellular functions?

Author Response

Reviewer #1

Comments

Comment 1: It is important to test if addition of antioxidants to PFE would improve the parameters after freezing-thowing to the level of GFE? Moreover, how these parameters in GFE is affected by removing the antioxidants from the seminal plasma?

Answer: Thanks for your question. The addition of enzymatic and non-enzymatic antioxidants to sperm freezing medium has been extensively studied in various species. However, the results shown by these studies are contradictory, because, as seen in this and other manuscripts, although the addition of antioxidants can improve sperm freezability, the most important thing is to maintain the oxidant/antioxidant balance. In addition, as could be observed in our study, there is a great interaction between the different enzymatic and non-enzymatic antioxidants, with which it is clear that several molecules with antioxidant action intervene in the regulation of oxidative stress caused by ROS.

In this sense, we can mention that our group previously carried out a study (https://doi.org/10.3390/vetsci8120302), in which we added different concentrations of reduced glutathione (GSH) to the freezing medium in donkey sperm, a species phylogenetically close to the horse. Although we did not observe a substantial improvement in sperm motility and functionality parameters, we did observe an effect on ROS control after thawing with the addition of high concentrations (8 mM and 10 mM) of GSH. In this case, we saw a resistance of donkey sperm to the toxicity produced by high concentrations of GSH, which is not observed in the horse, considering concentrations above 2.5 mM of GSH as cytotoxic for sperm (https://doi.org/10.1016/j.jevs.2013.05.001).

We believe that more studies should be carried out that include different antioxidant molecules and different concentrations of these added to the freezing medium to try to improve sperm freezing, also taking into account that the imbalance of the oxidant/antioxidant balance in favor of antioxidants also can be detrimental to sperm quality.

On the other hand, the removal of seminal plasma before sperm freezing is performed routinely in horses, because it has been seen that some seminal plasma proteins are detrimental to semen cryopreservation. However, in recent years the composition and importance of seminal plasma in various species has been more thoroughly studied. Studies that have shown that some components of seminal plasma with antioxidant properties are enhancers for sperm cryotolerance. In equids it has been observed that the brief contact between sperm and seminal plasma before its elimination could be enough for seminal plasma antioxidants to exert an improved effect on sperm cryotolerance (lines 464-466 of the discussion). At this point it is also important to mention that the seminal plasma is not completely eliminated (100%), since a small portion is always left behind, where studies indicate that it leaves a small amount of seminal plasma (approximately 5%) better results to freezing (https://doi.org/10.1111/j.1439-0531.2010.01729.x) and modulation of post-AI uterine inflammatory response (https://doi.org/10.1038/s41598-021 -99972-9). So, in the case of GFE, which show good sperm cryotolerance, in terms of total motility and viability (which has been shown to be related to the antioxidants present in seminal plasma), if a large amount of plasma is left seminal in the sperm to be frozen or it is not removed, on the contrary its cryotolerance would be affected.

Comment 2: How extracellular enzymatic antioxidants affect intracellular functions?

Answer: Thanks for your question. The imbalance between the production of reactive oxygen species (ROS) and the ability of biological systems to easily detoxify reactive intermediates or easily repair the resulting damage is known as oxidative stress (https://doi.org/10.1016/S0015-0282(02)04948-8). In the specific case of the sperm, this is a cell that is particularly susceptible to suffering oxidative stress caused by ROS, due to the large amount of polyunsaturated fatty acids (PUFA) present in the plasma membrane and the low cytoplasmic volume. This condition causes the intracellular production of antioxidants by the sperm to be limited, acquiring a dependence on the antioxidants present in the seminal plasma.

In this sense, extracellular enzymatic antioxidants protect sperm by controlling ROS (for example, in the case of PON1 by controlling lipoperoxides) and thus avoiding oxidative stress, which can cause intracellular damage to sperm such as lipid peroxidation. The effects on the intracellular function of extracellular enzymatic antioxidants in sperm can be observed in this study in the positive correlation between enzymatic antioxidants with the percentage of sperm with an intact plasma membrane and acrosome, as well as with sperm motility. This indicates that the effect of antioxidants allows the sperm to maintain its structure and functionality. At this point it is important to mention that similar effects were obtained in a recent study by our group in frozen-thawed donkey semen (https://doi.org/10.3390/antiox11020417), where a positive correlation of some antioxidants could be observed enzymatic and non-enzymatic with the percentage of sperm with an intact plasma membrane and motility, in addition to a negative correlation with the percentage of sperm with sperm membrane lipid disorder.

Reviewer 2 Report

The main issue discussed in this paper is comprehensive research on the influence of plasma enzymatic and non-enzymatic antioxidant systems on the susceptibility of stallion sperm to cryo-damage in the process of sperm preservation at low temperatures. Interestingly, the levels of antioxidant activity in seminal plasma, and in thawed semen, the kinetic, morphological and physiological characteristics of the sperm of stallions with good and poor freezing abilities were analyzed.

I believe that the topic is original and appropriate in this field because the cryopreservation of stallions semen, is still a serious challenge for science. And each new study, and in particular the role of enzymatic (such as paoaoxonase type -1 PON1 and superoxide dismutase - SOD) and non enzymatic (Trolox equivalent anitioxidant capacity - TEAC) in cryotolerance in the stallion semen, may contribute to the increase in knowledge and even to obtain breakthrough results in the future.

An important advantage of the work is that the authors quantified, for the first time in horses, the activity levels of PON1 in seminal plasma, and they evaluated its relationship with sperm cryo tolerance. Furthermore, authors stated that this enzyme plus SOD could play a key role as an antioxidants in horse semen. Also they found for the first time that TEAC in seminal plasma  was positively related to post-thaw sperm plasma membrane integrity. This work fills a serious gap in the knowledge of the influence of the stallion’s plasma antioxidants on cryo-damage in the sperm, which may be associated with a future increase in the usage of frozen semen in horse reproduction.

In my opinion the whole manuscript is outstanding, the descriptions of the materials and methods are very clear and good, the presentation of the results in the form of heat maps and box-whisker plots is excellent.

The conclusions are in line with the evidence and arguments presented and relate well to the main question posed.

The references presented are adequate, they were selected with care and knowledge of the subject. The references are very current and well chosen.

Due to the high value of the authors' work, I have allowed myself to leave no comments regarding this manuscript.

Author Response

Reviewer #2

Comments

General comment: In my opinion the whole manuscript is outstanding, the descriptions of the materials and methods are very clear and good, the presentation of the results in the form of heat maps and box-whisker plots is excellent.

Due to the high value of the authors’ work, I have allowed myself to leave no comments regarding this manuscript.

Answer: Thank you for your comment. We really appreciate your positive feedback and would like to thank you for your kind review of our manuscript. This significantly motivates our group to continue conducting scientific research in this area.

Round 2

Reviewer 1 Report

None